# Feasibility and acceptability of a dietary intervention study to reduce salt intake and increase high-nitrate vegetable consumption among middle-aged and older Malaysian adults with elevated blood pressure: a study protocol

Andrea McGrattan [1], Devi Mohan [2,3] Pei Wei Chua,[2,3] Azizah Mat Hussin,[4] Yee Chang Soh,[2,3] Mawada Alawad,[2,3] Zaid bin Kassim,[5] Ahmad Nizal bin Mohd Ghazali,[5] Blossom Stephan,[6] Pascale Allotey,[2] Daniel D Reidpath,[3,7] Louise Robinson [1], Mario Siervo [8]

For numbered affiliations see end of article.

**Correspondence to**
Dr Andrea McGrattan;
Andrea.McGrattan@Newcastle.ac.uk

## ABSTRACT

**Introduction** Global population ageing is one of the key factors linked to the projected rise of dementia incidence. Hence, there is a clear need to identify strategies to overcome this expected health burden and have a meaningful impact on populations' health worldwide. Current evidence supports the role of modifiable dietary and lifestyle risk factors in reducing the risk of dementia. In South-East Asia, changes in eating and lifestyle patterns under the influence of westernised habits have resulted in significant increases in the prevalence of metabolic, cardiovascular and neurodegenerative non-communicable diseases (NCDs). Low vegetable consumption and high sodium intake have been identified as key contributors to the increased prevalence of NCDs in these countries. Therefore, nutritional and lifestyle strategies targeting these dietary risk factors are warranted. The overall objective of this randomised feasibility trial is to demonstrate the acceptability of a dietary intervention to increase the consumption of high-nitrate green leafy vegetables and reduce salt intake over 6 months among Malaysian adults with raised blood pressure.

**Methods and analysis** Primary outcomes focus on feasibility measures of recruitment, retention, implementation and acceptability of the intervention. Secondary outcomes will include blood pressure, cognitive function, body composition and physical function (including muscle strength and gait speed). Adherence to the dietary intervention will be assessed through collection of biological samples, 24-hour recall and Food Frequency Questionnaire. A subgroup of participants will also complete postintervention focus groups to further explore the feasibility considerations of executing a larger trial, the ability of these individuals to make dietary changes and the barriers and facilitators associated with implementing these changes.

**Ethics and dissemination** Ethical approval has been obtained from Monash University Human Research Ethics Committee and Medical Research and Ethics Committee

## Strengths and limitations of this study

► Feasibility of the first 2×2 trial testing single and combined effects of dietary nitrate and salt reduction for the prevention of cognitive decline in developing countries.

► Longest high-nitrate-based intervention ever conducted.

► Use of objective biomarkers for the assessment of adherence to the dietary intervention and internal validation of the dietary assessment methods used (urinary sodium—24-hour urine vs spot urine/nitrate levels in blood—dried blood spots vs plasma and ELISA vs gold standard/nitrate in saliva—salivary strip (point of care test) vs ELISA using saliva samples).

► Mixed-method approach to obtain key information on the feasibility of the dietary intervention.

► Owing to the small sample size and the feasibility nature of this study, the planned comparative analysis is only exploratory, and therefore, the efficacy of the intervention between groups cannot be determined. However, the results of this study will be instrumental in the design and calculation of the sample size of a larger, follow-on efficacy trial.

of Malaysia. Results of the study will be disseminated via peer-reviewed publications and presentations at national and international conferences.

**TRIAL REGISTRATION NUMBER**
ISRCTN47562685; Pre-results.

## INTRODUCTION

Dementia is characterised by multiple cognitive deficits and loss of independence, and its pathogenesis is linked to multiple distinct

neuropathological processes including Alzheimer's disease and/or vascular pathology.[1] Several of the pathological processes underlying dementia might be delayed, or prevented, by interventions focused on dietary and behavioural changes[2] and, in the absence of definite pharmacological treatments, these options represent a key strategy to alleviate its individual and social impact.[3] Optimal control of modifiable cardiovascular factors, such as blood pressure, and adherence to lifestyle recommendations have been consistently associated with reduced dementia risk in observational studies.[2–4] The role of nutrition for dementia prevention has been extensively investigated by testing the role of single nutrients (ie, n-3 Poly Unsaturated Fatty Acids or B vitamins[5]) or dietary patterns (ie, dietary approaches to stop hypertension (DASH) diet[6] or Mediterranean diet[7 8]). There are a small number of randomised controlled trials (RCTs) available to support the effect of diet on cognition in mild cognitive impairment[9] and among those at risk of cognitive decline or cognitively healthy populations[10]; however, the trials have a small sample size and overall modest quality.

Malaysia is experiencing rapid socioeconomic and nutritional transitions.[11] Changes in individual eating and lifestyle patterns and rise in food availability and financial prosperity of the population have resulted in significant increases in the prevalence of chronic metabolic, cardiovascular and neurodegenerative non-communicable diseases (NCDs).[12] Effective public health interventions for the prevention of dementia are challenging in countries with limited social and healthcare resources which are unable to meet the growing demands of ageing populations. A systematic review of 22 RCTs investigated the effect of nutritional interventions on cognitive performance in developing economies of East Asia, and the majority of the studies showed some significant beneficial effects of the nutritional interventions on cognitive performance. However, these effects were not consistent across all of the neuropsychological tests used in each study.[10] Therefore, there is a need for well-designed nutritional intervention studies for the early prevention of cognitive decline and dementia.

Recent analyses have identified inorganic nitrate as a potential nutrient improving vascular and metabolic health.[13] Beetroot and green leafy vegetables, such as spinach, lettuce and rocket, are a key source of naturally available inorganic nitrate, containing over 250 mg of nitrate per 100 g of fresh vegetable weight.[13] Inorganic nitrate is closely linked to the metabolism of nitric oxide (NO) which is known for its multiple effects on physiological functions such blood pressure, brain function and immunity.[14] An increase in inorganic nitrate consumption has been associated with an increase in NO production via complex mechanisms involving oral microbiota, gastric environment and activation of reducing enzymes.[15] Recent epidemiological studies and clinical trials have demonstrated the association of increased nitrate intake with improved blood pressure, metabolic health and cognitive function.[14 16] A 15-year prospective cohort study investigated the association of consumption of high-nitrate vegetables and risk of atherosclerotic vascular disease (ASVD) mortality. Results showed that a consumption of approximately 30 mg/day higher nitrate intake from vegetables was associated with a 21% lower risk of ASVD mortality. This nitrate amount equates to approximately 10–30 g/day of nitrate-rich vegetables.[17] However, the impact on cognition is still unknown with studies of short intervention durations and among small samples, and limited studies conducted in developing countries.[13 14] Nevertheless, with hypertension highlighted as an important modifiable risk factor for dementia,[3 18] more studies that test the implementation of dietary interventions that aim to improve blood pressure are warranted.

Sodium is the main electrolyte contained in extracellular fluids and involved in regulation of blood pressure and cellular activities. An excess sodium intake has been linked to extracellular volume expansion and increased blood pressure.[19] The DASH diet has been specifically developed to promote consumption of healthy dietary choices alongside strictly controlled sodium intake.[20] Secondary analyses derived from seminal DASH studies showed additional benefits on blood pressure and cardiometabolic health in individuals with lower salt intake and adherence to a healthy dietary pattern.[21] These analyses have been extended to cognitive function with promising results,[22 23] but the evidence has mostly come from studies conducted in the USA and no studies to our knowledge have been conducted in low/middle-income countries (LMICs). Epidemiological studies have repeatedly proven the association of high salt diets with impaired blood pressure control and increased cardiovascular risk[24 25]; this evidence has been confirmed in several clinical trials testing the effects of salt-reducing interventions and demonstrating the protective effects of salt reduction on cardiovascular health.[26 27] A prospective study looked at the relationship between a reduction in salt intake with blood pressure, mortality from stroke and ischaemic heart disease (IHD).[28] A decrease in salt intake by 1.4 g/day, as measured by 24-hour urinary sodium, contributed to a reduction in stroke and IHD mortality, and in addition, among individuals who were not on antihypertensive medication, there was a significant fall in systolic blood pressure (SBP) of 2.7±0.34 mm Hg.[28] Excessive sodium consumption is recognised as a global health issue[29] with efforts aiming to reduce salt intake at the population level by 30% by 2025. Seventy-five countries have already implemented national sodium reduction programmes to help meet this target.[30]

In sum, an increased consumption of dietary nitrate and lower salt intake may have additional benefits on health outcomes such as blood pressure and cognition, mediated by an increase in NO generation (ie, nitrate) and control of vascular tone and cellular metabolism (ie, sodium). However, this hypothesis has never been tested in previous interventions which represents the main aim of the feasibility study proposed in this protocol.

The Global Health Dementia Prevention and Enhanced Care (DePEC) project is a National Institute for Health Research (NIHR) funded study with a key goal to develop approaches for dementia prevention and develop more efficient postdiagnostic care in LMICs. The DePEC-Nutrition feasibility study (presented here) is one of five work streams within the overall DePEC project.

## Purpose and aims

The overall objective of the DePEC-Nutrition feasibility study is to demonstrate the acceptability of a dietary intervention to increase the consumption in high-nitrate green leafy vegetables and reduce salt intake over a 6-month period among Malaysian adults with raised blood pressure. The objectives focus on the core areas of a trial that need to work for the study to succeed procedurally, such as recruitment ability and participant retention, data collection procedures and assessment methods used, potential to deliver the dietary intervention and resource requirement. Information on the effect size of the intervention on cognition and blood pressure will also be determined, which will be instrumental in the design and calculation of the sample size of a follow-on efficacy trial.

## METHODS AND ANALYSIS
### Study design

The DePEC-Nutrition feasibility study is a 6-month randomised 2×2 factorial trial including four parallel arms: (1) control, (2) high-nitrate vegetable consumption, (3) low salt consumption and (4) combined high-nitrate vegetable consumption plus low salt consumption. The trial will be conducted among 120 middle-aged and older Malaysian adults. Sponsor and grant number: GHR Group: 16/137/62—NIHR Global Health Research Group on DePEC, Newcastle University, UK. The Standard Protocol Items: Recommendations for Interventional Trials reporting guidelines were used to guide the preparation of this protocol.[31]

## Participants

This study is embedded within the South East Asia Community Observatory (SEACO) population (http://www.seaco.asia/). SEACO is a health and demographic surveillance site (HDSS); a unique research platform in population health and well-being focused on a middle-income community in Segamat, Johor in the southern tip of the Peninsular Malaysia. The district of Segamat comprises urban, rural and plantation areas with an ethnic mix including Malays, Chinese and Indian, in similar proportions to the national population. SEACO runs its HDSS research platform in five subdistricts (mukim) of the Segamat district: Bekok, Chaah, Gemereh, Jabi and Sungai Segamat. The SEACO population was established using a baseline census (March 2012 to February 2013) that included information on sociodemographic status and self-reported health conditions. Consenting participants underwent detailed health profile assessments in the first health round in 2013–2014. The next census was done in 2017, with a total number of 40 015 participants completing the assessment. Individuals were seen again after 1 year (in 2018) for the second health round. Individuals aged 50 years and over, who are living in Segamat, Johor and have been assessed as part of SEACO during the health round in 2018 will form the study population for the DePEC-Nutrition feasibility study. The baseline assessment will be carried out in the Klink Kesihatans (KK) in Sungai Segamat subdistrict (a community health clinic in Segamat, Johor). Hence, only participants from within 5 km radius of KK Segamat will be included in the study population. A total of 2433 participants have been identified on the database as potentially eligible for this study by broad application of the study criteria (mean age (SD): 61.27 (6.77) years and number of men: n=1078 (44.3%)). This subset of participants will be screened in detail, and then will be informed of the study and invited for a screening visit as outlined in the study procedures.

## Recruitment

The community will be informed about this study via a local Community Engagement Committee who meet regularly to distribute information on projects planned within SEACO. Participants will be randomly identified by the application of the inclusion and exclusion criteria to the SEACO health surveillance database. Potential participants will be approached via a home visit and will be provided with information about the study. They will be given an opportunity to read detailed information about the study procedures and can ask questions. Thereafter, their interest of participating in the study will be evaluated. If they agree to take part, a screening assessment will be completed. If eligible, participants will be invited to attend a baseline study visit at the health facilities within the community.

## Eligibility criteria

To be eligible for inclusion in this study, participants must report as:
1. Male and female participants with an age between 50 and 75 years from the SEACO database.
2. Pre-hypertensive, stage 1 hypertensive or diagnosed hypertensive: any person with a self-reported history of hypertension (on or not on medication) or with elevated blood pressure (SBP 120–159 mm Hg or diastolic blood pressure (DBP) 80–99 mm Hg).
3. Not severely cognitively impaired (defined as a Mini Mental Examination Status (MMSE) Score of 19 or over).

Participants will be excluded from the study if they:
1. Are participating in other research clinical studies.
2. Are unable to provide consent.
3. Have SBP more than or equal to 160 mm Hg and/or DBP more than or equal to 100 mm Hg.
4. Adhere to any therapeutic diet such as weight loss treatments or a gluten-free diet.

5. Are vegan (these individuals are likely to have a high intake of vegetables). Since Indian participants are more likely to follow a vegetarian or semi-vegetarian diet, in order to maintain representativeness across ethnic groups included in the study, only participants who are vegan will be excluded from the study.
6. Participants with a body mass index (BMI) less than $18.5 \, \text{kg/m}^2$.
7. Have a history of active cancer or any diagnosis of cancer in the last 5 years.
8. Have a history of excessive alcohol intake (>21 units of alcohol per week).
9. Have a history of brain damage or significant head trauma (resulting in a loss of consciousness).
10. Have a diagnosis of acute and chronic medical conditions interfering with the study outcomes such as systemic infections (tuberculosis, hepatitis B, HIV/AIDS), severe liver and kidney diseases, inflammatory bowel diseases, coronary heart diseases, cerebrovascular diseases or is a diabetic on insulin therapy.
11. Have had any major surgical procedures (in the past 6 weeks or planned in the next year) that could interfere with the study outcomes.
12. Have a current diagnosis of moderate or severe depression, or other serious mental or brain disorder and/or are currently on psychiatric medication (anti-depressants, sedatives, antipsychotics).
13. Regularly use sodium-altering drugs (ACE inhibitors, angiotensin receptor blockers, corticosteroids, diuretics, hormonal therapies (oestrogens, thyroxin and progesterone), non-steroidal anti-inflammatory drugs, proton-pump inhibitors), organic nitrates.
14. Have changed their antihypertensive medication regimen in the previous 3 months.
15. Have limited mobility due to any reason.
16. Are planning to move house within a year.

### Study procedure

Data collection will be undertaken directly on an electronic hand held device (Samsung Galaxy Tab 3v—Survey CTO). Questionnaire items will be read out loud by the data collector, and participant responses will be recorded directly onto the electronic tablet. The flow of participants through the DePEC-Nutrition feasibility study is illustrated in figure 1. At the screening visit, written informed consent will be first obtained, after which participants will complete a screening questionnaire to collect information on demographics, comorbidities, medication history, cognitive function (MMSE), body weight, height and resting blood pressure. If eligible, participants will undergo baseline assessments, consisting of two parts: a home visit (part 1), followed by an appointment at KK, Segamat (part 2). Baseline assessments will include: obtaining respondent's signed informed consent for study participation, eligibility assessment (medical and medication history), dietary assessment (24-hour recall and Food Frequency Questionnaire (FFQ)), cognitive assessment, collection of biological samples (blood, dried

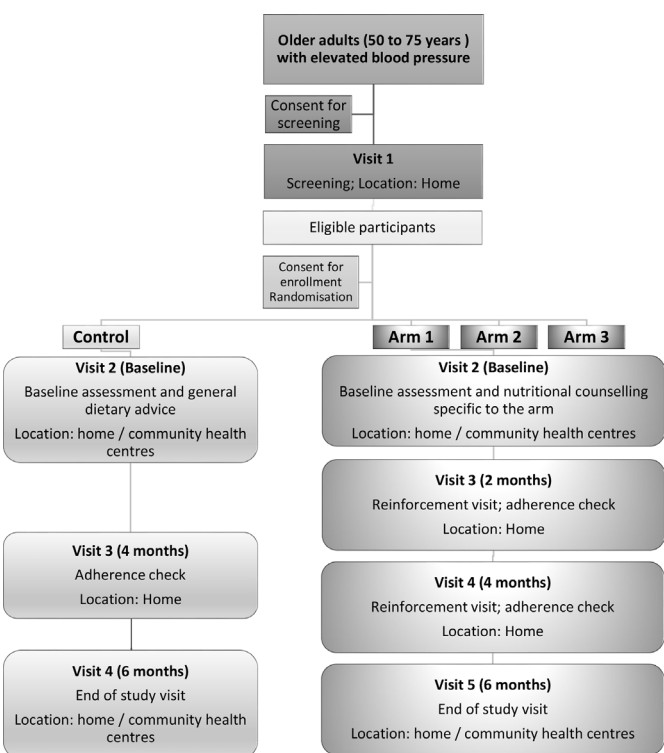

**Figure 1** A flow diagram providing an overview of the study procedures.

blood spots, 24-hour urine and spot urine samples, saliva samples and salivary strips), measurements of hand grip and gait speed, physical activity (International Physical Activity Questionnaire (IPAQ)) and depression (Geriatric Depression Scale (GDS)).

An interim 2-month visit will take place for those in the intervention groups only, to provide reinforcement messages relating to the allocated dietary intervention and with a view to enhancing compliance with the intervention. In addition, a brief adherence questionnaire will be administered, body weight and resting blood pressure will be measured and salivary strip and spot urine samples will be collected. The same measurements and further dietary reinforcement messages will be conducted at an interim 4-month visit for the participants allocated to the intervention arms. Participants in the control arm will only have one interim visit at month 4 and will have their weight and blood pressure measured. The end of study visit (at KK, Segamat) will include the conduction of the measurements collected at baseline, including cognitive assessment, dietary assessment and collection of biological samples (blood, dried blood spots, 24-hour urine and spot urine samples, saliva samples, salivary strips). In addition, an exit group session (focus group evaluation) and a self-administered feedback questionnaire will be scheduled to evaluate the adherence to the interventions and obtain detailed feedback on the overall study protocol. Finally, the participants will be offered a token of appreciation to acknowledge their participation in the study and to compensate for travel expenses. This is usually to the value of RM 25.

## Randomisation and blinding

The SEACO health round database will serve as the sampling frame, from which eligible participants are identified. Block randomisation will be carried out to assign eligible participants into one of the four arms using R software (randomizeR).[32] Block sizes of four participants will be used in the randomisation and will be generated by a member of the research team not involved in the data collection. The interviewers will be blinded to the participant assignment to intervention arms. Participant blinding to the intervention allocation will not be possible due to the nature of the dietary intervention study. Matching of characteristics for group allocation will not be used in this study. The distribution of characteristics, such as age or gender, will be part of the feasibility assessment to gain information on the characteristics of the population willing to participate without imposing any restrictions on the intention to participate in the study. This information will be used to evaluate whether specific randomisation approaches will need to be used in future studies to ensure even distribution of key characteristics between groups.

## Intervention and control arms

This dietary and behavioural intervention will focus on key components that are related to changing local dietary patterns influenced by nutritional transition trends in Malaysia. It aims to target two key components that have protective (dietary nitrate) and negative (salt intake) effects on cardiovascular and cognitive health. The intervention will target individuals with a higher Cardiovascular Disease (CVD) risk to adopt a tailored strategy to enhance daily consumption of green leafy vegetables and/or reduce salt intake. Nutritional counselling sessions with a medical doctor will be conducted at baseline to provide sustainable and effective strategies to improve dietary habits as well as monitor and address key barriers to dietary change. These sessions will be delivered in a group setting using PowerPoint slides and practical activities to educate the group on the allocated dietary intervention. All intervention groups will receive written information focused on health benefits, sources, recommended intakes and practical tips related to one of the following dietary interventions: (1) to increase dietary nitrate, (2) to reduce salt intake and (3) to increase dietary nitrate and to reduce salt intake. Those who are randomised to the salt intervention (either alone or combined) will receive a measuring spoon as an additional strategy to support individuals to understand portion size and measure salt intake, as successfully tested by an intervention study in China.[33] Furthermore, the Ministry for Health Malaysia set out in their Salt Reduction Strategy 2015–2020 how reducing the amount of salt and sauces consumed in the home is a key priority.[34] This provides justification for inclusion of the salt restriction spoon as an intervention strategy. The intervention groups will also receive bi-weekly text message reminders to provide reinforcement of the key dietary behaviour change messages relevant for their group allocation. Compliance will be monitored during the intermediate visits (2 and 4 months, respectively) where participants will also receive reinforcement video messages to remind participants of the key dietary advice that was discussed during the counselling sessions at baseline. Those allocated to the control group (4) will receive written information on general healthy eating and lifestyle advice based on the Malaysian dietary guidelines.[35]

## Outcome measures

Outcome assessments will be conducted at baseline, month 4 and month 6 for all groups, with an additional assessment visit at month 2 for those in the intervention groups. Assessments at baseline and month 6 will be conducted in clinic/at home, with remaining assessments completed in the participants own home. The same instruments will be used for measurements at each time point. Assessors are trained prior to the assessments to follow a standardised protocol. A timeline of the study procedures can be found in table 1.

### Primary outcome

The primary outcome measures are related to feasibility and will be collected via questionnaire, review of the study procedures and by review of data collector field notes. Patient feedback, acceptance and adherence issues will be additionally assessed through qualitative focus groups conducted post intervention. The specific feasibility considerations that will be assessed in the study have been outlined in table 2. The CONSORT 2010 statement[36] will be used to guide the reporting of the feasibility results.

### Secondary outcomes
#### Cognitive assessment

A validated version of Montreal Cognitive Assessment test, auditory verbal learning test, categorical verbal fluency and trail making test part B will be conducted at baseline and end of study visits.

#### Blood pressure

Three consecutive measurements of resting blood pressure readings will be recorded in a sitting position using a calibrated OMRON automated monitor (OMRON HEM 907, OMRON Healthcare, Milton Keynes, UK). The median measure will be used to estimate the blood pressure. Blood pressure will be measured at all outcome assessment visits.

#### Anthropometry and body composition

Body weight will be recorded in kilograms using calibrated, electronic scales to the nearest 0.1 kg where possible. Weight will be measured at all outcome assessment visits. Height will be recorded in metres using a stadiometer and measured to the nearest 0.01 m where possible. The recordings for weight and height will be subsequently used to calculate BMI by dividing weight (kg) by height ($m^2$). BMI will be calculated at baseline and end of study visits. Body composition will be measured

**Table 1** Timeline of study procedures

| Procedures | Consent/screening | Baseline Week 1 | Week 2 | Week 4 | Week 6 | Week 8 | Week 10 | Week 12 | Week 16 | Week 18 | Week 20 | Week 22 | End of study Week 24 |
|---|---|---|---|---|---|---|---|---|---|---|---|---|---|
| Inclusion/exclusion | ✓ | | | | | | | | | | | | |
| Consent | ✓ | | | | | | | | | | | | |
| **Activities for intervention group** | | | | | | | | | | | | | |
| Dietary education | | ✓ | | | | | | | | | | | |
| Reinforcement at home visit | | ✓ | | | | ✓ | | | ✓ | | | | |
| Group education counselling session | | ✓ | | | | | | | | | | | |
| Home visit | | ✓ | | | | ✓ | | | ✓ | | | | |
| SMS messages | | | ✓ | ✓ | ✓ | ✓ | ✓ | ✓ | ✓ | ✓ | ✓ | ✓ | |
| **Activities for control group** | | | | | | | | | | | | | |
| General health promotion message | | ✓ | | | | | | | | | | | |
| Home visit | | ✓ | | | | | | | ✓ | | | | |
| **Study measurements** | | | | | | | | | | | | | |
| Sociodemographic | | ✓ | | | | | | | | | | | |
| Concomitant medication check list | | ✓ | | | | ✓ | | | | | | | ✓ |
| Any medical events | | ✓ | | | | ✓ | | | | | | | ✓ |
| Physical activity (IPAQ) | | ✓ | | | | | | | | | | | ✓ |
| Depression (GDS) | | ✓ | | | | | | | | | | | ✓ |
| Food Frequency Questionnaire (FFQ) | | ✓ | | | | | | | | | | | ✓ |
| 24-hour dietary recall | | ✓ | | | | ✓† | | | ✓ | | | | ✓ |
| Nitrate: urine | | ✓* | | | | ✓†‡ | | | ✓‡ | | | | ✓* |
| Salt: urine | | ✓* | | | | ✓†‡ | | | ✓‡ | | | | ✓* |
| Cognitive performance | | ✓ | | | | | | | | | | | ✓ |
| Resting blood pressure | | ✓ | | | | ✓† | | | ✓ | | | | ✓ |
| Body composition | | ✓ | | | | ✓†§ | | | ✓§ | | | | ✓ |

Continued

**Table 1** Continued

| Procedures | Consent/screening | Baseline Week 1 | Week 2 | Week 4 | Week 6 | Week 8 | Week 10 | Week 12 | Week 16 | Week 18 | Week 20 | Week 22 | End of study Week 24 |
|---|---|---|---|---|---|---|---|---|---|---|---|---|---|
| Physical function | | ✓ | | | | | | | | | | | ✓ |
| Blood sample and dried blood spot | | ✓ | | | | | | | | | | | ✓ |
| Saliva sample and salivary strips | | ✓ | | | | ✓†¶ | | | ✓¶ | | | | ✓ |

*Twenty-four hour urine and spot urine samples.
†Intervention groups only.
‡Spot urine sample only.
§Body weight only.
¶Saliva strips only.
FFQ, Food Frequency Questionnaire; GDS, Geriatric Depression Scale; IPAQ, International Physical Activity Questionnaire.

by bioelectrical impedance. A Tanita Body Composition analyser will be used to measure body fat percentage and classification, segmental subcutaneous fat and skeletal muscle percentage (whole body, trunk, legs and arms), resting metabolism, visceral fat level and classification and body age. Body composition will be measured at baseline and end of study visits.

### Physical assessments

The following physical assessments will be conducted at baseline and end of study visits. (1) Muscle strength will be measured by a hand grip-strength dynamometer. Using their right arm, the participant will be asked to make three measurements and an average calculated. The same process will be repeated for the left arm. (2) A 4-metre walk test will be done to assess the gait speed. Two trials will be conducted and the average measure of time in minutes will be used to assess the gait speed. (3) In the Timed Up and Go (TUG) test, the time taken for participants to rise from an armless chair (46 cm height), to walk 3 m, turn, walk back and sit down will be measured. The TUG test will be performed two times consecutively, and the average of the two scores will be used. (4) The GDS will be used to assess depressive symptoms[37]—scores range from 0 to 15, with a score of 0 to 5 indicating a normal score and a score greater than 5 suggesting depression. The IPAQ[38] will be used to assess physical activity.

### Dietary assessment

Conventional approaches to assessing dietary intake are associated with measurement error. There is a growing consensus that combining the use of self-report instruments (like FFQ and 24-hour recall) together with biomarker analyses could increase the accuracy of individual intake estimates, especially for episodically consumed foods.[39 40] Thus, dietary assessment data will be collected by two methods: 24-hour recall and FFQ. Participants will complete a 1-day 24-hour recall facilitated by a trained data collector. The FFQ is a validated questionnaire used in previous dietary research in Malaysia.[41] Information will be entered into Nutritionist Pro Software (V.7.5) and daily intake will be calculated.

### Biological sample collection

The following bio-specimen will be collected in this study:
1. Whole venous blood (15 mL).
2. Twenty-four-hour urine sample.
3. Spot urine sample.
4. Saliva.
5. Capillary blood.

All samples will be used only for the purpose of the study. The venous blood will first be centrifuged, separated into aliquots and then stored at −20°C at the KK Segamat and then transported to Monash University Sunway campus for storage. Other samples will be stored at −20°C storage. The freezer room for the stored samples will be under lock and key with restricted access.

**Table 2** Primary outcome: feasibility considerations to be assessed

| Feasibility considerations | |
| --- | --- |
| Recruitment | 1. Recruitment rates<br>2. Time taken for the recruitment of participants<br>3. Characteristics of recruited sample as against expected<br>4. Non-response rates<br>5. Reasons for non-response |
| Retention and completion | 1. Retention rates<br>2. Number of drops and reasons for withdrawal<br>3. Follow-up response rates (2 months and 4 months follow-ups) |
| Feasibility of measurement tools | 1. Time taken to fill in questionnaires<br>2. Missing data from questionnaires<br>3. Understanding level of procedures and measurement protocols<br>4. Suitability of outcome measures<br>5. Sensitivity of outcome measures to change<br>6. Internal consistency of outcome measures—validation of data collection methods (24 hours vs spot urine; saliva samples vs salivary strips) |
| Resource capacity | 1. Data collection and measurement equipment<br>2. Field support (transportation, IT and so on)<br>3. Personnel (were the right numbers of people with the right skills available when they were needed)<br>4. Duration of each phase |
| Acceptability and adherence to intervention | 1. Barriers to participation (collected at screening interview)<br>2. Burden (reasons for not taking part/discontinuation or dropping out)<br>3. Qualitative enquiry/feedback from participants and research staff<br>4. Focus group evaluation—this part of the feasibility trial will separately examine the acceptability and suitability of: intervention materials; intervention timing; intervention procedures; measurement and sample collection; adherence to intervention and barriers and facilitators to dietary and lifestyle changes |

### Blood sampling

Whole blood samples will be analysed to measure changes in nitrite concentrations during the intervention. Samples will also be analysed to assess for biomarkers of cardiovascular risk (C reactive protein, glycated haemoglobin, nitro-tyrosine) and direct brain measures such as plasma brain-derived neurotrophic factor (BDNF), plasma amyloid β42 and amyloid β40. A capillary blood sample will be taken and analysed for the concentration of glucose in the blood using a portable glucometer. Dried blood spot samples will be analysed to measure changes in nitrate concentrations during the intervention. Blood spots will be first processed using a standardised elution protocol to obtain a liquid solution of the samples and then analysed for nitrate concentrations using ozone-based chemiluminescence, which is the reference method for the analysis on nitrate in biological fluids. All blood sampling assessment methods will be conducted at baseline and end of study visits. Venous blood samples will be performed by a qualified medical attendant or staff nurse practising at KK Segamat and dried blood spot samples will be collected by a trained data collector.

### Twenty-four hour urine and spot urine sampling

Both 24-hour urine and spot urine samples will be collected. Twenty-four-hour urine collection has been associated with a having high respondent burden owing to the time consuming nature of this method, particularly in a community and LMIC setting, where 24-hour urine sampling may be logistically difficult. Spot urine sampling is potentially a more convenient and affordable alternative. However, there are still a number of questions about the reliability of spot urine collection as a means of monitoring intervention adherence. Therefore, by using these two methods of urine collection, the authors hope to explore the feasibility of implementing these collection methods within the target population.[42]

Eligible participants will be provided with a kit to collect a 24-hour urine sample during their baseline home visit (part 1). This will be provided 2 days before their clinic visit (part 2). The kit includes: (1) a participant information booklet with written instructions on how to collect the sample; (2) a '24-hour urine collection record' to note essential information about the urine collection; (3) urine-collecting equipment: (a) 2.5-litre screw-capped plastic container for storing the 24-hour urine sample, (b) 1-litre plastic jug which the urine will be voided into, (c) 1-litre screw-capped plastic container either as backup container when the 2.5-litre storage container is full, or for temporal collections of urine made outside the home and (4) a permanent marker pen to note the start and

finish times of urine collection in the 2.5-litre container. Trained data collectors will verbally explain the method of 24-hour urine collection to participants. The 24-hour urine collection will be initiated 1 day before their clinic visit and end in the morning of their appointment. Participants will also collect a spot urine sample on the morning of their clinic visit. They will be provided with a 60 mL screw-capped plastic bottle for the spot urine collection. Participants will be instructed by a trained data collector to provide a spot urine sample using the midstream clean-catch technique.

The 24-hour urine samples collected will be assessed for completeness using assessment of the duration of urine collection, the total urine volume and 24-hour urinary creatinine excretion. The urine samples will be excluded from analysis if the time of the collection falls outside the range of 22–26 hours, if the total 24-hour urine volume is less than 500 mL or greater than 6000 mL and if 24-hour creatinine excretion is less than 3 mmol or greater than 25 mmol in women, or less than 6 mmol or greater than 30 mmol in men. Urinary sodium will be determined using the ion-selective electrode method. For nitrate, samples will be diluted 1:100 and then analysed for nitrate concentrations using ozone-based chemiluminescence, which is the reference method for the analysis on nitrate in biological fluids.

### Saliva sampling

Whole saliva will be collected using the passive drool technique. One millilitre of which will be collected into one collection tube. Participants will be asked to generate some saliva in their mouth and when ready, the participant should hold the adaptor with collection tube attached to their mouth and pass saliva into the tube. This process is repeated until desired amount is collected. To measure nitrate, samples will be diluted 1:100 and then analysed for nitrate concentrations using ozone-based chemiluminescence. For the salivary strips, a dedicated app freely available to download on mobile devices will be used to provide a quantitative reading of the salivary nitrite concentrations as a surrogate marker of dietary nitrate intake.

### Patient and public involvement—post intervention qualitative evaluation

Participants will be invited to provide feedback on their participation in the intervention study via focus groups discussions. The focus group topic guide will also be piloted among a random sample of participants prior to implementation to allow feedback, refinement and tailoring prior to implementation. Two focus group discussions will be conducted within each arm of trial (total 8). Purposive sampling will be used to capture broad variation in age, ethnicity, house location and for better understanding of the phenomenon being studied. All focus groups discussions will be audio recorded and the topics covered will include: (1) access to the food items—financial (affordability), physical factors; (2) cultural acceptability of the

recommended food items; (3) family support to adhere to the dietary changes; (4) adherence to the instructions—facilitators and barriers, methods to self-monitor; (5) perceived benefit and side effects of intervention; (6) study context—education material, frequency of visit, location of visits, data collection methods and tools; (7) provision of biomedical samples—barriers, cultural acceptability, difficulty; (8) likes and dislikes in relation to the new dietary options and (9) preferred intervention arm. Finally, a behaviour change theoretical framework will be used to guide the qualitative analysis.[43 44] This will allow the feedback and recommendations from participants to directly inform refinements and tailoring of the intervention for a follow-on efficacy trial as well as to understand the barriers and facilitators to dietary change among this target group.

### Sample size

A formal sample size calculation has not been performed considering the pilot, feasibility nature of the study. A sample size of 30 per group was based on the estimated effect size provided by the guidelines by Whitehead *et al*,[45] which outlines the sample size calculation for pilot studies with the aim to maximise resources and avoid occurrence of a type II error. Specifically, a sample size of greater than 25 individuals per group would provide in a 90% powered main trial the ability to detect a small effect size between 0.1 and 0.3. More than 100 participants are expected to complete the trial; that is, an anticipated drop-out rate ≤20%.

### Statistical analysis

As the main aims of this feasibility study relate to the feasibility, acceptability and the potential to deliver a dietary intervention, these data will be reported narratively illustrated with descriptive statistics. The CONSORT 2010 statement will be used to guide the reporting of this information.[36] Secondary aims are largely descriptive, aiming to provide bounds for key parameters to inform the main trial. Intention-to-treat analysis will be used to include all randomised participants and determine the key outcome measures. The recruitment and retention rate in each arm will be described. It is unlikely that statistically significant differences in retention rates will be discovered, but upper and lower bounds will be estimated and reasons for groups with apparently low recruitment or retention rates will be explored qualitatively. Normality of the distribution of the variables and appropriate transformations (LogX, 1/X, Xn) will be performed if necessary. Summary data will be expressed as mean (SD) or frequency (%). General linear models for repeated measures will be used to detect significant differences between the intervention groups with and without adjustment for baseline levels. $\chi^2$ test will be used for categorical variables. An interaction term (time×group) will be built to assess between-group interactions in changes in the measured outcomes during the interventions. Dietary data will be analysed using the Nutritionist Pro Software (V.7.5).

## Data collection supervision and training

SEACO is an ISO-certified research platform for its operations, therefore training and supervision are carried out in accordance with SEACO's standard operating procedures. All data collectors are required to be fluent in the local language (Bahasa Melayu or Malay language or Mandarin) and have at least a working knowledge of English. To ensure data quality and consistency across interviewers, the field supervisor will undertake random concurrent supervisory visits to observe practice. The data collectors will also be periodically observed by the project leader and the field manager. Based on the scoring completed during these observation sessions, onsite training or re-training will be carried out as appropriate. Data collection is undertaken on Samsung Galaxy tablets. The data are encrypted on the tablet to ensure they are secure and inaccessible in the unlikely event that they are mislaid by data collectors. Transfer of data from the tablet to the remote server is done on a weekly basis. The data are also encrypted on the server and backed up by servers at Monash Sunway Campus and at Monash Clayton. Research staff have limited access to anonymised data.

## Data monitoring

All expected and unexpected adverse events reported by participants will be recorded in an events register and reported to the Human Research Ethics Committee. Due to the nature of the treatment products, that is, commercially available food products, no adverse events are expected. However, if participants feel in anyway adversely affected by any foods or the principal investigator feels an adverse event necessitates cessation, the participant would be advised not to continue and the appropriate measures will be taken (ie, record in field notes, contact research nurse and principal investigator if deemed necessary).

## Ethics and dissemination

The procedures outlined in this protocol, pertaining to the conduct, evaluation and documentation of this study are designed to ensure that the sponsor and investigator abide by Good Clinical Practice (GCP) Guidelines and under the guiding principles detailed in the Declaration of Helsinki. DM and YCS have obtained GCP certification from Ministry of Health, Malaysia. Ethical approval has been granted by Monash University Human Research Ethics Committee (17864) and the Malaysian Medical Research Ethics Committee (#NMRR-19-617-45916). Results of the study will be disseminated via peer-reviewed publications and presentations at national and international conferences.

## DISCUSSION

This study will demonstrate the feasibility of a dietary intervention to increase the consumption of high-nitrate green leafy vegetables and reduce salt intake over a 6-month period among Malaysian adults with raised blood pressure. The design of this feasibility study has taken into account the evidence collected in developed countries on the efficacy of complex interventions for the prevention of cognitive decline and evidence on the major transitions in dietary patterns occurring in developing countries as a result of nutrition transition trends. An increased consumption of vegetables and reduction of salt intake are key dietary recommendations for the prevention of cardiometabolic diseases. Our hypotheses are that a targeted, educational approach, to encourage healthy dietary choices focused on these key elements of the diet, would be linked to greater long-term adherence to the interventions and improved cognitive function with prospective, projected reduced risk of dementia in these populations. This mixed-method feasibility study will provide key quantitative and qualitative information on the delivery of a novel dietary intervention and will aim to estimate the effect size of the single and combined interventions on cognitive function and blood pressure, which will be instrumental in the design and calculation of the sample size of a larger, follow-on efficacy trial.

**Author affiliations**

[1]Population Health Sciences Institute, Newcastle University, Newcastle upon Tyne, UK

[2]Global Public Health. Jeffrey Cheah School of Medicine and Health Sciences, Monash University Malaysia, 47500 Subang Jaya, Selangor, Malaysia

[3]South East Asia Community Observatory (SEACO), Monash University Malaysia, Segamat, Johor, Malaysia

[4]Kampus Cawangan Institute of Medical Science Technology, Universiti Kuala Lumpur, Kuala Lumpur, Wilayah Persekutuan, Malaysia

[5]District Health Office, Pejabat Kesihatan Daerah (PKD) Segamat, Segamat, Johor, Malaysia

[6]School of Medicine, University of Nottingham, Nottingham, UK

[7]International Centre for Diarrhoeal Disease Research, ICDDR,B, Dhaka, Bangladesh

[8]School of Life Sciences, University of Nottingham, Nottingham, UK

**Contributors** MS, DDR and DM were involved in conception and trial design. MS, AM and AMH contributed to the intervention development and design. AM wrote the initial manuscript draft and MS and DM contributed to writing the final manuscript and provided critical comments during revisions. All authors contributed to a critical review of the paper (AM, DM, PWC, AMH, YCS, MA, ZBK, ANBMG, BS, PA, DDR, LR and MS). DM and PWC will be responsible for overseeing the recruitment, data collection and intervention delivery.

**Funding** This study is funded by the National Institute for Health Research (NIHR) [GHR Group: 16/137/62—NIHR Global Health Research Group on Dementia Prevention and Enhanced Care (DePEC), Newcastle University, UK using UK aid from the UK Government to support global health research. The views expressed are those of the author(s) and not necessarily those of the NIHR or the Department of Health and Social Care. Sponsor: Monash University, Malaysia Jeffrey Cheah School of Medicine & Health Sciences, Selangor 47500, Malaysia. Tel: +60 (0) 7 2190 600/Population Health Sciences Institute; Newcastle University, Newcastle, UK].

**Competing interests** LR reports grants from National Institute of Health Research Senior Investigator award during the conduct of the study; no other relationships or activities that could appear to have influenced the submitted work.

**Patient and public involvement** Patients and/or the public were not involved in the design, or conduct, or reporting, or dissemination plans of this research.

**Patient consent for publication** Not required.

**Provenance and peer review** Not commissioned; externally peer reviewed.

## ORCID iDs

Andrea McGrattan http://orcid.org/0000-0003-1521-213X
Devi Mohan http://orcid.org/0000-0002-0898-2729
Louise Robinson http://orcid.org/0000-0003-0209-2503
Mario Siervo http://orcid.org/0000-0001-5515-0944

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
