## [Reviewer comments · BMJ Open]

ARTICLE DETAILS

TITLE (PROVISIONAL)	The feasibility and acceptability of a dietary intervention study to reduce salt intake and increase high-nitrate vegetable consumption among middle-aged and older Malaysian adults with elevated blood pressure: a study protocol
AUTHORS	McGrattan, Andrea; Mohan, Devi; Chua, Pei Wei; Mat Hussin, Azizah; Soh, Yee Chang; Alawad, Mawada; bin Kassim, Zaid; bin Mohd Ghazali, Ahmad Nizal; Stephan, Blossom; Allotey, Pascale; Reidpath, Daniel; Robinson, Louise; Siervo, Mario

VERSION 1 – REVIEW

REVIEWER	Michihiro Satoh Tohoku Medical and Pharmaceutical University, Japan
REVIEW RETURNED	20-Jan-2020

GENERAL COMMENTS	Thank you for giving me the opportunity to review this article. This study will assess the feasibility of dietary intervention study and will also provide information for the calculation of sample size. It seems a pilot study, however, I have several comments on this protocol. 1. The introduction is too long. Furthermore, I understood the importance of dietary intervention but there is little information on the “feasibility” of dietary intervention in previous studies. The effect of excessive salt intake on health is general knowledge.2. The characteristics such as age or proportion of men in the study population should be shown.3. For recruitment, do they give the participants any incentives?4. What factors will be used to match the groups? For instance, gender will considerably affect the results.5. I believe the main intervention is dietary intervention at baseline and sending SMS messages during follow-up. However, the number of home visits and study measurements are different between the control and intervention groups. This will make the results complex.6. Is the primary outcome measured in standard methods? If these are new indices, it is difficult to compare it with other results.7. In the statistical method, will the authors perform an intention-to-treat analysis for each outcome?
---

REVIEWER	Jacqui Webster The George Institute for Global Health, Sydney, Australia
REVIEW RETURNED	24-Feb-2020

GENERAL COMMENTS	Overall -
-----------

	1. This is an important study and the outcomes could be extremely influential for policy and practice 2. One of the things that seems to be missing from this protocol and what seem to me to be the most important question for this feasibility study is what magnitude of the reduction in salt or increase in nitrates is required to demonstrate a change in cognitive status. I don't think it will be possible to move to a larger scale efficacy trial unless this question is addressed. Please could you make reference to this in the introduction and explain how you will address this in the methods. 3. This protocol requires thorough editing by someone with English as a first language. I've spent considerable time going through and suggesting edits but it would be helpful if one of the authors could also do a final edit of English language Specific comments, mainly editorial: Page 3 The title is confusing and I didn't realise it was a feasibility study until I got quite far into the paper. I suggest changing the title to: "Protocol for feasibility study of a dietary intervention trial to reduce salt Isn't "potential to deliver" part of feasibility? Or is it feasibility of the intervention and potential to undertake the trial? Line 29 should be increase consumption "of" Line 32 should read "Secondary outcomes will include ..." Line 34 – please explain "physical function" in this context Line 32-36 – it is not clear whether the clause "to assess adherence to the dietary intervention" applies to all of the secondary outcomes just listed or just to the biological samples. Please reword to clarify Line 38 – I suggest changing to "further explore the feasibility considerations of executing a larger trial" Page 4 Line 7 – should say "Feasibility study for first ever 2*2 trial" Line 16-17 Possibility of internal validation of dietary assessment methods? Possibility seems a bit vague for strengths. Please clarify. Line 19 – should be feasibility of intervention? Page 4 Introduction – at 5 pages, this is currently too long and is a bit repetitive and unstructured. I suggest shortening and re-structuring so that the information is presented in a logical way. For example, the results of the systematic review should be presented before the description of the project in the last paragraph Page 5 Line 30 – Should read "urgent need to identify protective risk factors ..." Lines 37 and 38, should read "high salt, fat and sugar" not refined sugar and salt. Lines 24 – 25 – now I realise this is a feasibility study for a bigger trial. That needs to be made clear from the start Line 39 – this doesn't currently make sense. Please reword for clarity Line 40 – should read "cardiovascular disease and type 2 diabetes" Line 43 – not clear what you mean by "such an approach". Please re-word to clarify what approach you mean Line 52 – I suggest ending the sentence after enzymes as the final part is not really needed/is potentially confusing Page 5
--	---

	Line 51 – Change “The results are still preliminary” To “However, the impact on cognition is still unknown with studies of short intervention durations and among small samples, and limited studies conducted in developing countries.” Line 66 – it is not clear what “these” relates to here. Maybe just delete Similarly and these and just say, “The healthy dietary patterns in these studies are characterised ...” Line 70. Change “a strict control of sodium intake” to “Strictly controlled sodium intake” Line 80 – interventions should be plural Line 82 – full stop and new sentence after 2025 Line 84 – delete the first “used” Line 84 – change “to include” to “including” Page 6 Line 88 – delete “therefore” Line 113 Change “This therefore provides rationale in support of the conduction of the pilot feasibility study presented in this protocol” To “This provides the rationale for the pilot feasibility study presented in this protocol”. Page 6 Purpose and aims Line 118 – again, just wondering whether potential to deliver is part of feasibility? Line 120 states” The objectives focus on the core areas of an intervention study that need to work for the study to succeed procedurally” It is not really clear what this means. Please reword to clarify. Page 7 Line 141 Change “close to national proportions” to “in similar proportions to the national population” Line 142 change to “established using a baseline ...” Line 144 – Change to “Consenting participants...” Line 145 – change to “assessments” Line 145 – change “the latest census” to “the next census” Line 146 – change “individual have been seen” to “individuals were seen” Line 147 Delete “which collected a detailed health profile of all the 148 consenting participants” as repetitive and implies the consenting only applies to the last census round when it should apply to all. Line 149 – not clear what previous refers to here. Please reword this section to clarify Page 8 Recruitment Line 156 - Please explain how you will inform the community Line 157 – please clarify how the random identification will occur Line 162-163 – please clarify what the baseline study visit consists of Please also include information about when and where you will ask participants to provide signed consent forms Eligibility criteria Line 167 Change “Male and female participants with an age between 50 and 75 years from the Line 177 Change “Any person who has a self-reported history of hypertension (on or not on medication) or those who had a blood pressure 171 recording of systolic blood pressure 120-159 mmHg or diastolic blood pressure 80-99 mmHg.” To “Any person with a self-reported history of hypertension (on or not on medication) or with
--	---

	elevated blood pressure (SBP 120-159 mmHg or DBP 80-99 mmHg.) Line 178 – I think it would make more sense to include the range of BP in the inclusion criteria 181-184 – Please change to Vegan (not strictly vegan) Also, I'm not sure I understand the rationale of excluding vegan but not vegetarian. Both groups will be eating similarly high levels of fruit and vegetables. The vegan group will just also be excluding dairy products and eggs. Vegetarian is different to vegan. A strictly vegetarian diet is not a vegan diet – they are different. Page 9 Line 188 – does this mean anyone who has ever lost consciousness would be excluded or does the loss of consciousness have to be linked to extreme trauma. Please clarify how you will define this Line 190-193 – Change “or diabetes on insulin therapy” to “or is a diabetic on insulin therapy” Line 195 – Change “or planned in next one year” to “or planned in the next year” Line 197 Should this read “and/or are currently on psychiatric medication (antidepressants, sedatives, antipsychotics)? Line 198 “Have systemic use of” doesn't make sense. Maybe “Is a systemic user of” or “regularly uses”? Line 202 – Change “Have a change in” to “has changed” Line 204 – Change “in next one year” to “within a year” Study procedure It would be helpful to have a flow chart for the study procedure Please add details about who will be collecting the data including their relevant training and qualifications to collect blood and urine samples Line 207 - Change “Data collection will be recorded directly on an” to “Data collection will be undertaken on an..” Line 2014. Suggest putting full stop after Visit 2. Then delete “and undergo their” and start a new sentence to read “Baseline assessments will include....” Line 2018 – Change “And with the aim to enhance compliance to the intervention” to “And with a view to enhancing compliance with the intervention.” Line 220 – should read “body weight and resting blood pressure will be measured” Visit 4 is the same as visit 3 for the intervention groups so just say this. It is not clear what the control group do on visit 3. Please explain Please provide further details of how you will collect 24 hour and spot urine samples including what equipment will be provided, methods for ensuring sample completeness, how samples will be collected, detailed instructions to participants etc. Maybe refer to WHO guidance on collecting 24 hour urine samples https://www.cdc.gov/nchs/data/nhanes/nhanes_13_14/24_Hour_Urine_Study_Procedures_Manual.pdf http://applications.emro.who.int/docs/EMROPub_2018_EN_17032.pdf?ua=1&ua=1
--	--

	Intervention and control arms Line 254 – This Chinese intervention was based on the fact that most salt is added during cooking and at the table – is this still applicable in Malaysia? Line 249-252 – it is well known that most campaigns to change behaviour are not sustainable. Please can you provide evidence to support the fact that these interventions will work and indicate whether a particular behaviour change theory or approach is being followed. This will be useful in trying to understand whether the intervention is implemented effectively. Please provide details of the group counselling sessions as listed in Table 1 Page 14 – secondary outcomes Line 298 – Physical assessments It is not clear to me why all of these assessments are being undertaken. Please can you explain relevance to the study and outcome measures somewhere. Dietary assessment Please explain why you are undertaking both a 24 hour dietary recall and an FFQ and how you will analyse the information? Line 343 – please explain why you are collecting both 24 hour and spot urine samples. Spot urine samples are probably not helpful in this context (they are only useful for obtaining estimates of population intake). It would probably make more sense to just collect 24-hour samples. Please provide further details of how you plan to do this Page 16- Lines 354-356 Delete: “Although participants will not be involved in the recruitment and conduct of this study” Amend “participants will be invited to provide feedback on their participation intervention study via focus groups discussions.” To “Participants will be invited to provide feedback on their participation in the intervention study via focus groups discussions” Line 366 Change “Likes and dislikes to the new dietary options” to “Likes and dislikes in relation to the new dietary options” Line 367 Change “Preference to the trial arm.” To “preferred intervention arm” Line 368 – Change “to directly inform the refinement of a follow-on efficacy trial.” To “to directly inform the follow-on efficacy trial.” Line 371-373 – This sentence is hard to understand – I suggest splitting into 2. Line 381 Change “The main aims from the feasibility study” to “The main aims of the feasibility study” Discussion
--	---

VERSION 1 – AUTHOR RESPONSE

Reviewer 1 comments	Author’s response
1. The introduction is too long. Furthermore, I understood the importance of dietary intervention but there is little information on the “feasibility” of dietary intervention in previous studies. The effect of excessive salt intake on health is general knowledge.	Thank you for the useful comment. We have revised the introduction and added relevant information in regards to existing evidence of studies within this area of research.
2. The characteristics such as age or proportion of men in the study population should be shown.	The SEACO health round data will be used as our sampling frame. 2433 participants have been identified on the system as being potentially eligible by broad application of our study inclusion and exclusion criteria to the database (Mean age (standard deviation) = 61.27 (6.77) years and Men – n=1078 (44.3%). These participants will be further screened in detail, then approached and screened for the study. We will fully report study population characteristics when publishing the findings of this research. For now, these details have been added to the manuscript, page 7.
3. For recruitment, do they give the participants any incentives?	The participants will be offered a token of appreciation to acknowledge their participation in the study and to compensate for travel expenses. This is usually to the value of RM 25. We have now described this in our study procedures section, please see page 9.
4. What factors will be used to match the groups? For instance, gender will considerably affect the results.	We have not used matching in this study. As this is a randomised controlled trial, it is expected that at the end of recruitment, characteristics such as gender etc. will be distributed similarly across the arms. In addition, the distribution of these characteristics such as age or gender will be part of the feasibility assessment to gain information on the characteristics of the population willing to participate without imposing any restrictions on the intention to participate into the study. This information will be used to evaluate whether specific randomisation approaches will have to be used in future studies to ensure even distribution of key characteristics between groups.
5. I believe the main intervention is dietary intervention at baseline and sending SMS messages during follow-up. However, the number of home visits and study measurements are different between the control and intervention groups. This will make the results complex.	The main comparisons will be made between the baseline and 6 month assessments, whereby all arms undergo the same assessments. The interim study visits are also part of dietary intervention to deliver reinforcement messages on salt / nitrate or combined intervention. Therefore, comparisons on dietary adherence can be made between interventions arms (who receive reinforcement messages

	during the visit) compared to control (who receive no reinforcement messages).
6. Is the primary outcome measured in standard methods? If these are new indices, it is difficult to compare it with other results.	These are not new indices – feasibility will be assessed via the usual parameters (recruitment, retention, attrition rates etc.) as described in our paper in Table 2. The CONSORT 2010 statement (Eldridge et al., 2016) will be used to guide the reporting of the feasibility results and we have now included this reference in our paper. Please see page 13.
7. In the statistical method, will the authors perform an intention-to-treat analysis for each outcome?	Owing to the feasibility nature of this trial, statistical analyses are primarily conducted to identify the retention/recruitment/attrition rates, which can be done by descriptive statistics. Also, calculation of the sample size for a definitive trial can be performed descriptively. For secondary outcomes, analysis will be by ITT and we have now described this in our manuscript, please see page 18.
Reviewer 2 comments	Author's response
1. This is an important study and the outcomes could be extremely influential for policy and practice	Thank you for the supportive comment.
2. One of the things that seems to be missing from this protocol and what seem to me to be the most important question for this feasibility study is what magnitude of the reduction in salt or increase in nitrates is required to demonstrate a change in cognitive status. I don't think it will be possible to move to a larger scale efficacy trial unless this question is addressed. Please could you make reference to this in the introduction and explain how you will address this in the methods.	This is an important comment. The identification of the minimal meaningful change for nutritional interventions to have an effect on health outcomes is paramount. However, this evidence is still limited for both salt and dietary nitrate and therefore we have not modelled the study based on these outcomes and we hope to achieve an increase in dietary nitrate intake of at least 100mg/day and a decrease of daily salt intake of at least 1gram/day. Key references to support these targets are provided in the introduction. We have updated the introduction with evidence supporting these nutritional targets and further clarify the aims of our study. The methods have not been updated as study was not modelled or designed around minimal changes because of the limited existing evidence linking dietary nitrate and salt intake with health outcomes relevant to our study. Please refer to the introduction section, pages 4-5.
3. This protocol requires thorough editing by someone with English as a first language. I've spent considerable time going through and suggesting edits but it would be helpful if one of the authors could also do a final edit of English language	Thank you for the comment. We are slightly surprised as the paper has been written and edited by several authors with English as first language and/or with extensive academic and publishing experience. Nonetheless, we have carefully edited the manuscript to improve readability.
Page 3 The title is confusing and I didn't realise it	Thank you for the comment. We have amended the title to further emphasize the feasibility design of the study. It now

was a feasibility study until I got quite far into the paper. I suggest changing the title to: "Protocol for feasibility study of a dietary intervention trial to reduce salt Isn't "potential to deliver" part of feasibility? Or is it feasibility of the intervention and potential to undertake the trial?	reads, "The feasibility and acceptability of a dietary intervention to reduce salt intake and increase high-nitrate vegetable consumption among middle-aged and older Malaysian adults with elevated blood pressure: a study protocol".
Line 29 should be increase consumption "of"	Amended in text, please now see page 2.
Line 32 should read "Secondary outcomes will include ..."	Amended in text, please now see page 2.
Line 34 – please explain "physical function" in this context	By physical function we mean assessments of hand grip, gait speed and timed up and go. This has been amended in text, please now see page 2, which now reads, "physical function (including muscle strength and gait speed).
Line 32-36 – it is not clear whether the clause "to assess adherence to the dietary intervention" applies to all of the secondary outcomes just listed or just to the biological samples. Please reword to clarify	This has been amended in text to describe how the adherence to the dietary intervention will be assessed. Please now see page 2, which now reads, "Adherence to the dietary intervention will be assessed through collection of biological samples, 24 hour recall and food frequency questionnaire".
Line 38 – I suggest changing to "further explore the feasibility considerations of executing a larger trial"	Amended in text, please now see page 2.
Page 4, Line 7 – should say "Feasibility study for first ever 2*2 trial"	Amended in text, please now see page 3.
Page 4, Line 16-17 Possibility of internal validation of dietary assessment methods? Possibility seems a bit vague for strengths. Please clarify.	Thank you for the useful comment. We agree and we have removed the word possibility as this is an objective of the study. Please see page 3, which now reads, "Use of objective biomarkers for the assessment of adherence to dietary intervention and internal validation of dietary assessment methods used (Urinary sodium – 24 hr urine vs spot urine / Nitrate levels in blood - dried blood spots vs plasma, and ELISA vs gold standard / Nitrate in saliva – salivary strip (point of care test) vs ELISA using saliva samples)"
Page 4, Line 19 – should be feasibility of intervention?	Amended in text, please now see page 3.
Introduction – at 5 pages, this is currently too long and is a bit repetitive and unstructured. I suggest shortening and re-	The introduction section has been reduced and restructured as suggested.

structuring so that the information is presented in a logical way. For example, the results of the systematic review should be presented before the description of the project in the last paragraph	
Page 5, Line 30 – Should read “urgent need to identify protective risk factors ...”	Due to restructuring of the introduction as recommended by reviewers, this has now been omitted from the manuscript.
Page 5, Lines 37 and 38, should read “high salt, fat and sugar” not refined sugar and salt.	Due to restructuring of the introduction as recommended by reviewers, this has now been omitted from the manuscript.
Page 5, Lines 24 – 25 – now I realise this is a feasibility study for a bigger trial. That needs to be made clear from the start	We hope that this is now clearer from our title and emphasis on feasibility throughout the paper.
Page 5, Line 39 – this doesn’t currently make sense. Please reword for clarity	Due to restructuring of the introduction as recommended by reviewers, this has now been omitted from the manuscript.
Page 5, Line 40 – should read “cardiovascular disease and type 2 diabetes”	Due to restructuring of the introduction as recommended by reviewers, this has now been omitted from the manuscript.
Page 5, Line 43 – not clear what you mean by “such an approach”. Please re-word to clarify what approach you mean	Due to restructuring of the introduction as recommended by reviewers, this has now been omitted from the manuscript.
Page 5, Line 52 – I suggest ending the sentence after enzymes as the final part is not really needed/is potentially confusing	This has been amended as suggested. Please see page 4.
Page 5, Line 51 – Change “The results are still preliminary” To “However, the impact on cognition is still unknown with studies of short intervention durations and among small samples, and limited studies conducted in developing countries.”	This has been amended as suggested. Please see page 4-5.
Page 5, Line 66 – it is not clear what “these” relates to here. Maybe just delete Similarly and these and just say, “The healthy dietary patterns in these studies are characterised ...”	Due to restructuring of the introduction as recommended by reviewers, this has now been omitted from the manuscript.
Page 5, Line 70. Change “a strict control of sodium intake” to “Strictly controlled sodium intake”	This has been amended in text. Please see page 5.

Page 5, Line 80 – interventions should be plural	Amended. Please see page 5.
Page 5, Line 82 – full stop and new sentence after 2025	Amended. Please see page 5.
Page 5, Line 84 – delete the first “used”	Amended. Please see page 5.
Page 5, line 84 – change “to include” to “including”	Amended. Please see page 5.
Page 6, Line 88 – delete “therefore”	Due to restructuring of the introduction as recommended by reviewers, this has now been omitted from the manuscript.
Page 6, Line 113 Change “This therefore provides rationale in support of the conduction of the pilot feasibility study presented in this protocol” To “This provides the rationale for the pilot feasibility study presented in this protocol”.	Due to restructuring of the introduction as recommended by reviewers, this has now been omitted from the manuscript.
Page 6 Purpose and aims Line 118 – again, just wondering whether potential to deliver is part of feasibility?	Thank you for your comment. We have revised and rephrased where required to ensure clarity throughout the manuscript.
Page 6, Line 120 states” The objectives focus on the core areas of an intervention study that need to work for the study to succeed procedurally” It is not really clear what this means. Please reword to clarify.	This has been amended to read, “The objectives focus on the core areas of a trial that need to work for the study to succeed procedurally, such as recruitment ability and participant retention, data collection procedures and assessment methods used, potential to deliver the dietary intervention and resource requirement. Information on the effect size of the intervention on cognition and blood pressure will also be determined, which will be instrumental in the design and calculation of the sample size of a follow-on efficacy trial”. Please see page 6.
Page 7 Line 141 Change “close to national proportions” to “in similar proportions to the national population”	Amended. Please see page 7.
Page 7, Line 142 change to “established using a baseline ...”	Amended. Please see page 7.
Page 7, Line 144 – Change to “Consenting participants...”	Amended. Please see page 7.
Page 7, Line 145 – change to “assessments”	Amended. Please see page 7.
Page 7, Line 145 – change “the latest census” to “the next census”	Amended. Please see page 7.

Page 7, Line 146 – change “individual have been seen” to “individuals were seen”	Amended. Please see page 7.
Page 7, Line 147 Delete “which collected a detailed health profile of all the consenting participants” as repetitive and implies the consenting only applies to the last census round when it should apply to all.	This has been deleted. Please see page 7.
Page 7, Line 149 – not clear what previous refers to here. Please reword this section to clarify	Previous relates to the previous health round (the health round that was conducted in 2018). This has been updated in the text. Please see page 7.
Page 8 Recruitment Line 156 - Please explain how you will inform the community	SEACO conducts community engagements that involve a Community Engagement Committee (CEC). It is a platform established to distribute information on projects planned within SEACO – this is how the participants for this study will be informed. This has been incorporated into the manuscript, page 7.
Line 157 – please clarify how the random identification will occur	The SEACO health round data will serve as the sampling frame, from which eligible participants are identified. Block randomisation will be carried out to assign eligible participants into one of the four arms. Block sizes of four participants will be used in randomisation and will be generated by a member of the research team not involved in the data collection. Please see page 9.
Line 162-163 – please clarify what the baseline study visit consists of	Baseline assessments will include: obtaining respondent's signed informed consent for study participation, eligibility assessment (medical and medication history), dietary assessment (24-hour recall and food frequency questionnaire (FFQ)), cognitive assessment, collection of biological samples (blood, dried blood spots, 24-urine and spot urine samples, saliva samples, and salivary strips), measurements of hand grip and gait speed, physical activity (IPAQ) and depression (GDS). Please see page 9.
Please also include information about when and where you will ask participants to provide signed consent forms	Participants provide signed informed consent for study participation at the baseline study assessment visit. Please see the study procedures section.
Eligibility criteria Line 167 Change “Male and female participants with an age between 50 and 75 years from the	Amended to read, “Male and female participants with an age between 50 and 75 years from the SEACO database”. Please see page 7.
Line 177 Change “Any person who has a self-reported history of hypertension (on or not on medication) or those who had a blood pressure 171 recording of systolic blood pressure 120-159 mmHg or diastolic blood pressure 80-99 mmHg.” To “Any	Amended as suggested. Please see page 7.

person with a self-reported history of hypertension (on or not on medication) or with elevated blood pressure (SBP 120-159 mmHg or DBP 80-99 mmHg.)	
Line 178 – I think it would make more sense to include the range of BP in the inclusion criteria	Please see point 2 of the inclusion criteria, page 7.
181-184 – Please change to Vegan (not strictly vegan) Also, I'm not sure I understand the rationale of excluding vegan but not vegetarian. Both groups will be eating similarly high levels of fruit and vegetables. The vegan group will just also be excluding dairy products and eggs. Vegetarian is different to vegan. A strictly vegetarian diet is not a vegan diet – they are different.	Thank you for the useful comment. We have already provided a rationale for the inclusion of vegetarians in the sample as a strategy to minimise differences between ethnic groups. The reason for excluding vegans was related to the possible higher intake in vegetables and lower sodium intake in this group. These differences were observed by Clarys et al., 2014 (https://www.ncbi.nlm.nih.gov/pmc/articles/PMC3967195/) who showed a different fibre and sodium intake between these two groups. We have removed the word strictly vegetarian as suggested. Please see page 8
Page 9 Line 188 – does this mean anyone who has ever lost consciousness would be excluded or does the loss of consciousness have to be linked to extreme trauma. Please clarify how you will define this	The loss of consciousness is indicative of the more severe form of brain damage or head trauma, which will be excluded from this study. This will differentiate it from minor traumatic brain injury / mild concussion. This has been updated in the manuscript, page 8.
Line 190-193 – Change “or diabetes on insulin therapy” to “or is a diabetic on insulin therapy”	Amended as suggested. Please see page 8.
Line 195 – Change “or planned in next one year” to “or planned in the next year”	Amended as suggested. Please see page 8.
Line 197 Should this read “and/or are currently on psychiatric medication (antidepressants, sedatives, antipsychotics)?	Amended as suggested. Please see page 8.
Line 198 “Have systemic use of” doesn't make sense. Maybe “Is a systemic user of” or “regularly uses”?	Amended to read as “Regularly uses sodium-altering drugs”. Please see page 8.
Line 202 – Change “Have a change in” to “has changed”	Amended as suggested. Please see page 8.
Line 204 – Change “in next one year” to “within a year”	Amended as suggested. Please see page 8.
Study procedure It would be helpful to have a flow chart for the study procedure	Please see figure 1 which provides an overview of the study procedures, starting from screening visits to the end of study visits for each arm.

Please add details about who will be collecting the data including their relevant training and qualifications to collect blood and urine samples	1. Venous blood sampling - will be performed by qualified medical attendant or staff nurse practising at KK Segamat. Please see page 16. 2. Dried blood spot - will be obtained by trained data collector (DC). The DC will be trained by a PhD student who has a Medical Bioscience qualification. Please see page 16 2. 24-hour urine and spot urine - will be obtained by trained DC. The DC will be trained by a PhD student with a health science qualification. Please see page 16.
Line 207 - Change “Data collection will be recorded directly on an” to “Data collection will be undertaken on an..”	Amended as suggested. Please see page 8.
Line 2014. Suggest putting full stop after Visit 2. Then delete “and undergo their” and start a new sentence to read “Baseline assessments will include...”	Amended. Please see page 9.
Line 218 – Change “And with the aim to enhance compliance to the intervention” to “And with a view to enhancing compliance with the intervention.”	Amended. Please see page 9.
Line 220 – should read “body weight and resting blood pressure will be measured”	Amended. Please see page 9.
Visit 4 is the same as visit 3 for the intervention groups so just say this.	Amended. “The same measurements and further dietary reinforcement messages will be conducted at an interim four-month visit for the participants allocated to the intervention arms.” Please see page 9.
It is not clear what the control group do on visit 3. Please explain	No visit at 2 months for the control groups. Please see figure 1.
Please provide further details of how you will collect 24 hour and spot urine samples including what equipment will be provided, methods for ensuring sample completeness, how samples will be collected, detailed instructions to participants etc. Maybe refer to WHO guidance on collecting 24 hour urine samples	Eligible participants will be provided with a kit to collect a 24-hour urine sample during their baseline home visit (part 1). This will be provided two days before their clinic visit (part 2). The kit includes: (1) A participant information booklet with written instructions on how to collect the sample; (2) A '24-hour urine collection record' to note essential information about the urine collection; (3) Urine-collecting equipment: (a) 2.5 litre screw-capped plastic container for storing the 24-hour urine sample, (b) 1 litre plastic jug which the urine will be voided into, (c) 1 litre screw-capped plastic container either as backup container when the 2.5 litre storage container is full, or for temporal collections of urine made outside the home; (4) A permanent marker pen to note the start and finish times of urine collection in the 2.5 litre container. Trained data collectors will verbally explain the method of 24-hour urine collection to participants. The 24-hour urine collection will be initiated one day before their

	clinic visit and end in the morning of their appointment. Participants will also collect a spot urine sample on the morning of their clinic visit. They will be provided with a 60 mL screw-capped plastic bottle for the spot urine collection. Participants will be instructed by a trained data collector to provide a spot urine sample using midstream clean-catch technique. The 24-hour urine samples collected will be assessed for completeness using assessment of the duration of urine collection, the total urine volume and 24-hour urinary creatinine excretion. The urine samples will be excluded from analysis if the time of the collection falls outside the range of 22–26 hours, if the total 24-hour urine volume is less than 500 mL or greater than 6000 mL, and if 24-hour creatinine excretion is less than 3 mmol, or greater than 25 mmol in women, or less than 6 mmol, or greater than 30 mmol in men. Page 16.
Intervention and control arms Line 254 – This Chinese intervention was based on the fact that most salt is added during cooking and at the table – is this still applicable in Malaysia?	Yes – our inclusion of this tool is based on the fact that salt and salty sauces are added during cooking and preparation of meals. Malaysian Ministry of Health have published a salt reduction strategy which outlines that reducing the amount of salt and sauces used in the home as a key priority: http://www.moh.gov.my/moh/resources/Penerbitan/Rujukan/Salt_reduction_strategy_FA_2015_-2020.pdf We have added this reference to our manuscript to add further justification. Please see page 10.
Line 249-252 – it is well known that most campaigns to change behaviour are not sustainable. Please can you provide evidence to support the fact that these interventions will work and indicate whether a particular behaviour change theory or approach is being followed. This will be useful in trying to understand whether the intervention is implemented effectively.	Thank you for your important comment and we completely agree. We have incorporated the use of behavioural change techniques into the intervention design, such as reinforcement (via SMS reminders and reinforcement messages at interim visits) / social influences (through group counselling sessions) / demonstration of behaviours (at group counselling sessions). However, a behavioural change framework or theory was not used to guide the design process for this study. We will evaluate the use of these techniques through the qualitative study post-intervention and this will allow further tailoring and refinement of the intervention for a definitive trial. We would like to apply a behavioural framework (COM-B model / behaviour change wheel / theoretical domains framework, Michie et al. 2011; Atkins et al., 2017) to the analysis of the qualitative work. This will allow the feedback and recommendations from participants to directly inform refinements and tailoring of the intervention for a follow-on efficacy trial as well as to understand the barriers and facilitators to dietary change among this population group. We have added this to our manuscript – please see page 17.

Please provide details of the group counselling sessions as listed in Table 1	This is discussed in the intervention and control arms section on page 10. Please see the amendment on page 10, which describes the format of these sessions.
Page 14 – secondary outcomes Line 298 – Physical assessments It is not clear to me why all of these assessments are being undertaken. Please can you explain relevance to the study and outcome measures somewhere.	Thank you for the comment. We had several discussions in terms of including these measurements but in the end we decided that they would be useful to evaluate the feasibility of the protocols within the trial as to provide information on whether these measures could be used in future trials to stratify subjects based on measures of frailty or physical disability.
Dietary assessment Please explain why you are undertaking both a 24 hour dietary recall and an FFQ and how you will analyse the information?	We have updated this to read, “Conventional approaches to assessing dietary intake are associated with measurement error. There is a growing consensus that combining the use of self-report instruments (like FFQ and 24 hour recall) with biomarker analyses, could increase the accuracy of individual intake estimates, especially for episodically consumed foods.” See page 15.
Line 343 – please explain why you are collecting both 24 hour and spot urine samples. Spot urine samples are probably not helpful in this context (they are only useful for obtaining estimates of population intake). It would probably make more sense to just collected 24-hour samples. Please provide further details of how you plan to do this	This has been explained in the text, “Both 24 hour urine and spot urine samples will be collected. 24 hour urine collection has been associated with a having high respondent burden owing to the time consuming nature of this method, particularly in a community setting and LMIC, where 24-hour urine sampling may be logistically difficult. Spot urine sampling is potentially a more convenient and affordable alternative. However, there are still a number of questions about the reliability of spot urine collections as a means of monitoring intervention adherence for both nitrate and salt intake. Therefore, by using these two methods of urine collection, we hope to explore the feasibility of implementing these collection methods within the target population “. Please see page 16.
Page 16- Lines 354-356 Delete: “Although participants will not be involved in the recruitment and conduct of this study”	Deleted. Please see page 17.
Amend “participants will be invited to provide feedback on their participation intervention study via focus groups discussions.” To “Participants will be invited to provide feedback on their participation in the intervention study via focus groups discussions”	Amended. Please see page 17.

Line 366 Change “Likes and dislikes to the new dietary options” to “Likes and dislikes in relation to the new dietary options”	Amended. Please see page 17.
Line 367 Change “Preference to the trial arm.” To “preferred intervention arm”	Amended. Please see page 17.
Line 368 – Change “to directly inform the refinement of a follow-on efficacy trial.” To “to directly inform the follow-on efficacy trial.”	Amended. Please see page 17.
Line 371-373 – This sentence is hard to understand – I suggest splitting into 2.	Amended. Please see page 17.
Line 381 Change “The main aims from the feasibility study” to “The main aims of the feasibility study”	This has been updated to read, “As the main aims of this feasibility study relate to the feasibility, acceptability and the potential to deliver a dietary intervention, these data will be reported narratively illustrated with descriptive statistics. The CONSORT 2010 statement will be used to guide the reporting of this information”. Please see page 18
Discussion Line 430 – Change to “consumption of”	Amended. Please see page 19.

VERSION 2 – REVIEW

REVIEWER	Michihiro Satoh Tohoku Medical and Pharmaceutical University
REVIEW RETURNED	27-Apr-2020

GENERAL COMMENTS	Thank you for revising the manuscript along with my previous comments. The manuscript has been really improved. However, I have several further comments.  1. In the abstract, the objective of this study should be mentioned in the introduction. 2. I think the introduction is still too long. 3. Will the participants get the feedback of assessments just after each interim visit? 4. Even after the revision, the randomization method is still obscure. How do they randomize the participants? by computer or envelope method? What factors are considered when they decide the allocation. We usually consider age, the proportion of men, and other strong confounding factors affecting the outcomes for an allocation. 5. Now, I don't think they can catch the change of cognition among such a relatively young population (50-75) in half a year although it is not the main outcome. 6. Please mention the model number of blood pressure measurement devices they will use.
--

REVIEWER	Jacqui Webster The George Institute for Global Health, Australia
REVIEW RETURNED	28-May-2020

GENERAL COMMENTS	Dear authors, Congratulations on all your hard work and for your perseverance in responding to my extensive comments and addressing the queries of the other reviewers. I am happy to recommend this paper for publication. I look forward to seeing it in print and to seeing the results of the study in due course. Best wishes Jacqui
--

VERSION 2 – AUTHOR RESPONSE

Reviewer 1 comments	Author's response
1. In the abstract, the objective of this study should be mentioned in the introduction.	Many thanks for your comment. We have updated the abstract so that the objective is mentioned in the introduction section. Please see page 2.
2. I think the introduction is still too long.	Thank you for your comment. We have previously reduced the introduction section by approximately 400 words during the first revisions. We do believe that the content is important to set the context for the study, especially to discuss the mechanisms and evidence to support for the two dietary components included in the intervention. However, we have further reviewed and refined where possible.
3. Will the participants get the feedback of assessments just after each interim visit?	Participants do not receive specific feedback on assessments conducted at interim visits. If a serious adverse event occurred, then we would provide feedback as necessary. At these visits, those in the intervention arms will receive dietary reinforcement messages specific to their allocated intervention.
4. Even after the revision, the randomization method is still obscure. How do they randomize the participants? By computer or envelope method? What factors are considered when they decide the allocation? We usually consider age, the proportion of men, and other strong confounding factors affecting the outcomes for an allocation.	Thank you for your comment. Randomisation was computer generated using R software and this has been updated in the manuscript. We discussed matching in our previous revisions and mentioned how the distribution of characteristics such as age or gender will be part of the feasibility assessment to gain information on the characteristics of the population willing to participate without imposing any restrictions on the intention to participate into the study. This information will be used to evaluate whether specific randomisation approaches will have to be used in future studies to ensure even distribution of key characteristics between groups.

	We have added this explanation to the manuscript so that it is clear for readers. Please see pages 9-10.
5. Now, I don't think they can catch the change of cognition among such a relatively young population (50-75) in half a year although it is not the main outcome.	Yes we agree that a 6 month study duration is relatively short to see a change in cognition. However, our primary outcome for this study is feasibility, focussing on the core areas of a trial that need to work for the study to succeed procedurally, including the data collection procedures and assessment methods used. Thus, we hope to determine the feasibility of implementing cognitive assessments within this LMIC, community setting with the aim to inform a future efficacy trial with longer follow-up. In addition, we are hoping to see a possible trend in change in cognition which, even if not significant, will be highly informative for the sample size calculation of future larger trials.
6. Please mention the model number of blood pressure measurement devices they will use.	The model is OMRON HEM 907. This detail has been inserted into the manuscript. Please see page 14.

VERSION 3 – REVIEW

REVIEWER	Michihiro Satoh Faculty of Medicine, Tohoku Medical and Pharmaceutical University
REVIEW RETURNED	09-Jun-2020
GENERAL COMMENTS	Thank you for taking my comments into account. I have no further comments.